# The Coexistence of Nonalcoholic Fatty Liver Disease and Type 2 Diabetes Mellitus

**DOI:** 10.3390/jcm11051375

**Published:** 2022-03-02

**Authors:** Marcin Kosmalski, Sylwia Ziółkowska, Piotr Czarny, Janusz Szemraj, Tadeusz Pietras

**Affiliations:** 1Department of Clinical Pharmacology, Medical University of Lodz, 90-153 Lodz, Poland; tadeusz.pietras@umed.lodz.pl; 2Department of Medical Biochemistry, Medical University of Lodz, 92-215 Lodz, Poland; sylwia.ziolkowska@stud.umed.lodz.pl (S.Z.); piotr.czarny@umed.lodz.pl (P.C.); janusz.szemraj@umed.lodz.pl (J.S.)

**Keywords:** nonalcoholic fatty liver disease, type 2 diabetes mellitus, epidemiology, diagnosis, treatment, insulin resistance, metabolic syndrome, obesity

## Abstract

The incidence of nonalcoholic fatty liver disease (NAFLD) is growing worldwide. Epidemiological data suggest a strong relationship between NAFLD and T2DM. This is associated with common risk factors and pathogenesis, where obesity, insulin resistance and dyslipidemia play pivotal roles. Expanding knowledge on the coexistence of NAFLD and T2DM could not only protect against liver damage and glucotoxicity, but may also theoretically prevent the subsequent occurrence of other diseases, such as cancer and cardiovascular disorders, as well as influence morbidity and mortality rates. In everyday clinical practice, underestimation of this problem is still observed. NAFLD is not looked for in T2DM patients; on the contrary, diagnosis for glucose metabolism disturbances is usually not performed in patients with NAFLD. However, simple and cost-effective methods of detection of fatty liver in T2DM patients are still needed, especially in outpatient settings. The treatment of NAFLD, especially where it coexists with T2DM, consists mainly of lifestyle modification. It is also suggested that some drugs, including hypoglycemic agents, may be used to treat NAFLD. Therefore, the aim of this review is to detail current knowledge of NAFLD and T2DM comorbidity, its prevalence, common pathogenesis, diagnostic procedures, complications and treatment, with special attention to outpatient clinics.

## 1. Introduction

The term nonalcoholic fatty liver disease (NAFLD) appeared more than 40 years ago, but neither its pathophysiology nor diagnostic criteria were established for many years [1]. In NAFLD, the buildup of excessive fat in the liver is not triggered by significant alcohol consumption. The American Association for the Study of Liver Diseases (AASLD) Practice Guidance of 2018 categorizes NAFLD into nonalcoholic fatty liver (NAFL) or nonalcoholic steatohepatitis (NASH) [2]. NAFL can be diagnosed in cases of at least 5% hepatic steatosis without appearance of hepatocellular injury, while NASH is defined as at least 5% steatosis but with inflammation and hepatocyte injury (e.g., ballooning). In both histological types, fibrosis may or may not be observed [2]. In 2020, the term metabolic-associated fatty liver disease (MAFLD) was coined, which more appropriately characterizes the nature of the disorder; in the definition of MAFLD, the excessive consumption of alcohol is not an excluding criterion [3].

With respect to the causes of excessive accumulation of triglycerides (TG) in the liver, there are two forms of NAFLD: primary and secondary, which have the same clinical and histological signs. The latter occurs in patients with metabolic syndrome (MetS), especially coexisting with type 2 diabetes mellitus (T2DM) [4]. The former is less common and should be suspected in patients with hepatic steatosis without conventional risk factors (Table 1) [5,6].

It has been found that isolated ultrasonography (USS)-diagnosed NAFLD is not associated with increased mortality. However, in some cases it may progress to fibrosis, cirrhosis or hepatocellular carcinoma (HCC). It leads to increased liver-related and all-cause mortality, mainly due to cardiovascular causes, independent of other known factors [7]. Some data suggest that NAFLD is associated with damage to many organs and tissues, including the cardiovascular system [8].

Some data suggest that NAFLD may predict the development of T2DM in the future, and vice versa [9,10]. Additionally, the presence of NAFLD in diabetic patients is associated with poor glycemic control and other serious complications [11]. Similarly, the presence of T2DM in patients with excessive fatty liver infiltration can contribute to progress of NAFLD and damage of the organ [12].

To represent knowledge in the area of NAFLD and its association with T2DM, we searched PubMed papers focusing on common pathogenesis, diagnosis, complications and treatment. The following keywords were used: T2DM, insulin resistance, NAFLD, fatty liver, nonalcoholic fatty liver, nonalcoholic steatohepatitis, diagnosis, ultrasonography, transient elastography, complications, treatment, metformin, thiazolidinediones, GLP-1 agonists and DPP-4 inhibitors. The main aim of this review is to represent current knowledge on the coexistence of NAFLD and T2DM, its prevalence, common pathogenesis, diagnostic procedures and treatment, with special attention to outpatient clinics.

## 2. Epidemiology

The frequency of NAFLD differs in various parts of the world, but its prevalence is increasing worldwide [13]. Many studies suggest that its occurrence is determined by economic, genetic and ethnic risk factors [14]. Realistic assessment of the number of NAFLD cases is significantly affected by the diagnostic method used [15]. The problem with epidemiological data highlights the difficulty in distinguishing NAFLD from NASH, as well as the assessment of secondary causes of hepatic steatosis [16]. The proportion of patients with NAFLD in the world is 25.24% and is still increasing [17]. A higher prevalence is mostly observed in developed countries, with diets rich in saturated fats. For example, in Europe, NAFLD occurs in 23.17% of the population [18].

A meta-analysis published in 2021, comparing 32 studies, showed that the frequency of hepatic steatosis differs across populations, e.g., 31% in Asia [19]. Furthermore, data collected from the 2011–2014 National Health and Nutrition Examination Survey (NHANES) reported 21.9% (95% CI 20.6–23.3) cases of NAFLD among US adults [20].

Although the prevalence of NAFLD differs in various ethnic groups, it is more common among men and in patients older than 50 years [17]. Additionally, after adjustment for age, sex and race/ethnicity, it was found that NAFLD occurred significantly more often in overweight and obese individuals [21]. Similar observations have been found for other metabolic risk factors. A strong and independent correlation between NAFLD and insulin resistance (IR), in patients with or without diabetes, has been observed [22]. This problem also affects elderly patients. Data from the Rotterdam Study showed that NAFLD was present in 35.1% patients, but its prevalence decreased with advanced age [23].

NAFLD is an independent risk factor for MetS [24]. Furthermore, the criteria for MetS are similar to the symptoms of NAFLD [25]. Every component of MetS, as well as their number, increases the risk of NAFLD and its severity [26]. Furthermore, the great majority of NAFLD cases meet at least one MetS criterion, while 33% meet all five criteria [27]. Interestingly, in patients with MetS, fatty livers are diagnosed almost 10-fold more frequently than in patients without this disorder [28,29]. Nevertheless, T2DM is closely linked with MetS since it is considered to be a metabolic disorder. The key factor in the pathogenesis of NAFLD, T2DM and MetS is obesity. Although NAFLD can also occur in lean individuals, it has been found that the degree of steatosis increases with BMI [30]. Moreover, simple fatty liver will develop into NASH in 3% of non-obese patients, but in 20% of obese patients [31]. With respect to obesity and MetS, a European multi-cohort study of obese people found that MetS occurred in up to 78% of men and 65% of women [32]. Recent studies have suggested that MetS, obesity and T2DM may be associated with colonic diverticulosis [33,34]. Diverticula are structural changes in the wall of the colon that arise from a hernia of the colonic mucosa and submucosa as a result of defects in the muscle layers within the colonic wall [35]. Colonic diverticulosis is closely related to obesity. Excess visceral fat, which is important not only in obesity but also in NAFLD, T2DM, and MetS, is a significant risk factor in colonic diverticulosis. The development of the above diseases, as well as problems with gut microbiota, can contribute to the formation of diverticula [36]. It has been shown that the condition occurs more frequently in more severe cases of fatty liver. In patients with colonic diverticulosis and liver steatosis, hypertension, T2DM and hypothyroidism were observed more frequently. Moreover, fatty liver was more common in the more severe forms of colonic diverticulosis [37].

In some regions (e.g., the Asia-Pacific region), NAFLD is also diagnosed in patients with lower BMI [38]. In a Korean study of 29,994 adults, NAFLD was found in 12.6% of non-obese subjects and in 50.1% of obese subjects. Further, in non-obese NAFLD patients, especially women, significantly higher prevalence rates for other components of MetS than for obesity were observed [39]. Overall, studies show that NAFLD is found mostly in obese people, and the onset of hepatic steatosis in a lean person is a risk factor for expansion of fat mass [17,40]. An Indian study established a clinical association of NAFLD with dyslipidemia, hypertension and obesity. In this study, the T2DM population with these comorbidities had 38%, 17% and 14% higher risk for NAFLD, respectively. It should be emphasized that mean AST (aspartate aminotransferase) and ALT (alanine aminotransferase) levels in NAFLD patients were highest in those aged 25–40 and lowest in those aged 71–84 [41].

The prevalence of NAFLD in patients with abnormal glucose metabolism is very high, independently of diagnostic method. Ortiz-Lopez et al. found that patients with NAFLD had impaired fasting glucose (IFG) and/or impaired glucose tolerance (IGT) significantly more often than those without NAFLD [42]. Two major European studies reported NAFLD prevalence rates of 42.6–69.5% in large samples of T2DM patients [43]. A long term (5 years) Korean prospective cohort study, performed on 25,232 Korean men without T2DM, compared the incidence rate of T2DM according to the degree of NAFLD (normal, mild, and moderate to severe). The incidence of T2DM increased according to the degree of NAFLD (normal: 7.0%, mild: 9.8%, moderate to severe: 17.8%) [44]. The Ragama Health Study also revealed an increased risk of developing T2DM in patients with ultrasound-diagnosed NAFLD. Of 2984 subjects, 31% had NAFLD and 22.65% diabetes. After three years of observation, 19.7% patients with NAFLD and 10.5% without NAFLD developed T2DM. The frequency of diabetes was 64.2 and 34.0 per 1000 person-years for patients with and without NAFLD, respectively [9]. In a Polish study the frequency of ultrasound features of NAFLD in patients with newly diagnosed T2DM was 71% in patients with mean age 55.64 ± 13.42 years. Additionally, in patients with NAFLD, mean body weight, waist and hip circumferences, body mass index (BMI), liver enzyme activity, serum C-reactive protein, total cholesterol and TG were higher, while HDL-cholesterol was significantly lower. There were no statistical differences between parameters of glycemic control in groups with and without NAFLD [45]. On the other hand, Yan Y et al., by means of USS, found a high fatty liver (56.7%) in T2DM patients (69.6%) [46]. In the Edinburgh Type 2 Diabetes Study (ET2DS) (939 participants, aged 61–76 years with T2DM) ultrasound signs of fatty liver were present in 56.9% of participants. After excluding those with a secondary cause for steatosis, the prevalence of NAFLD in the study was 42.6%. Additionally, independent predictors of NAFLD were not only high BMI and TG, but also shorter duration of diabetes, HbA1c and metformin use [47]. In two meta-analyses from 2017 [48] and 2019 [49], 55.5–59.67% of T2DM cases had NAFLD, while 77.87% were simultaneously obese with fatty livers.

Despite the high prevalence of NAFLD and T2DM coexistence, AASLD guidelines do not recommend performing screening for NAFLD in diabetic patients. However, they suggest that screening tests for NAFLD are not cost–effective in relation to the long-term benefits [2].

## 3. Common Pathogenesis

The pathogenesis of T2DM and NAFLD is complex and not fully understood, but the presence of many common elements in the development of both diseases has been demonstrated. These include: alterations in glucose and lipid metabolism, IR, insulin secretion, genetic predisposition and environmental influences (such as endocrine disruptors), epigenetic factors and lifestyle changes [50]. Recently conducted studies indicate the complex interplay between carbohydrate metabolism and NAFLD and vice versa [51]. The development of NAFLD is strongly associated with hepatic IR. However, whether NAFLD is a consequence or cause of IR is a matter of debate. A number of studies conducted in NAFLD patients showed both an impaired ability of insulin to suppress endogenous glucose production (indicating the presence of hepatic IR) and approximately 50% reduction in glucose disposal (a measure of whole-body insulin-sensitivity) [52]. The associations between T2DM, NAFLD, IR are shown in Figure 1.

Chai et al. demonstrated that, in patients with similar levels of IR and hyperglycemia, NAFLD with T2DM was associated with higher serum insulin levels than T2DM alone. In these cases, hyperinsulinemia was caused mainly by beta-cell hypersecretion [53]. This observation was further confirmed in a study by Finucane et al., where healthy patients with NAFLD had higher insulin and C-peptide levels after oral glucose loading. However, in this group of patients, the C-peptide increment and adaptation index were significantly lower [54]. The association between NAFLD and IR can be explained by the “multiple-hit hypothesis”. The “first hit” is associated with impaired cellular response to insulin and compensatory IR. In adipose tissue, this affects hormone-sensitive lipase (HSL) increasing the risk of lipolysis with consequent release of free fatty acids (FFA) to the liver. Glucose absorption decreases in the skeletal muscles, while, in hepatocytes, hyperinsulinemia increases gluconeogenesis, decreases glycogen synthesis, increases uptake of FFA, alters the transport of TG, such as very low-density lipoproteins (VLDL), and inhibits beta-oxidation. These alterations in the metabolism of fat are the basis of NAFLD development.

The changes referred to above result from a complex interplay between various factors, such as hepatic resistance to leptin, or the reduction of adiponectin levels. The “second hit” is a consequence of oxidative stress in hepatocytes, which is initially compensated by cellular antioxidant mechanisms. However, overload of the liver with FFA generates reactive oxygen species (ROS) in the mitochondrial chain, which act on the fatty acids of the cell membranes, causing lipid peroxidation. ROS induce proinflammatory cytokine synthesis in Kupffer cells and hepatocytes. This second phase explains the evaluation necroinflammatory phenomenon, fibrosis and liver cirrhosis. There are many other “hits” involved in the pathogenesis of NAFLD, including genetic and epigenetic factors, the intestinal microflora and the cannabinoid system [55]. In the case of hepatic steatosis, T2DM or dysregulation of fasting glucose is found in 18–33% of NAFLD cases [56]. Moreover, these patients, as well as those with advanced fibrosis, have a higher IR [57]. Furthermore, patients with T2DM usually present with dyslipidemia with an elevated plasma TG, small LDL and decreased HDL cholesterol. IR results in lipolysis and, hence, increased circulating FFA, which are later taken up and oxidized by the liver, leading to formation of free radicals. TNF alpha, which is a proinflammatory marker, is also elevated in patients who are insulin resistant, as opposed to adiponectin, an anti-lipogenic and anti-inflammatory factor, which is decreased.

The pathogenesis of T2DM or NAFLD is similar to the processes observed in obesity. In this condition, elevated levels of very low-density lipoproteins (VLDL) can be observed which act as TG carriers and play a role in hypertriglyceridemia [58]. Moreover, increase in TG concentration is also caused by enhanced FFA flow to the liver and other tissues, which may contribute to the development of IR present in NAFLD, T2DM and obesity [59]. Additionally, the accumulation of fat in adipose tissue leads to the development of inflammation by stimulating the production of cytokines, e.g., TNF-α and IL-6, and may result in hypertension [60].

All the changes referred to above may lead to NAFLD and NASH. TNF alpha increases mitochondrial generation of ROS and recruits inflammatory cells, which can lead to steatohepatitis [61]. Moreover, the presence of the *PNPLA3* single nucleotide polymorphism, that is common in individuals with fatty livers, links hepatic steatosis and diabetes. *PNPLA3* encodes adiponutrin, which is associated with TG metabolism in adipocytes. Unfortunately, its function is poorly understood. The occurrence of the I148M (polymorphism in 148 position; isoleucine to methionine) variant increases the risk of T2DM and IR in NAFLD patients [62,63].

Nevertheless, the border between NAFLD and NASH is very thin. It is difficult to differentiate simple fatty liver from steatohepatitis without using invasive diagnostic methods. In a study of T2DM patients, each of the 63 research participants had fatty liver, but 94.82% of them had NASH, which was confirmed by liver biopsy. This suggests that steatohepatitis may be associated with the early stages of T2DM development, perhaps due to the association of NASH with IR [64].

One of the “hits” in the multi-hit hypothesis is the gut microbiota, which play a role not only in NAFLD, but also in T2DM. The gut microbiota is found to be an important regulator of the host’s energy metabolism [65,66]. Compounds are transported from the intestine to the liver through the portal vein, and the liver transfers antibodies back through the bile [67]. The permeability and composition of the mucus are dependent on the intestinal microflora, and its dysfunction results in the production of molecular patterns associated with pathogens (PAMP). Increased permeability results in enhanced liver inflammation [68,69]. Any changes that occur in its composition or its functioning may also affect metabolism in the liver, adipose tissue or muscles. Modulation of the intestinal microbiome may alleviate metabolic disorders by increasing the availability of fiber to microorganisms residing in the organism [70,71]. Moreover, less variability in the gut microflora was observed in NAFLD patients when compared to healthy controls [72,73].

Any disturbances in the gut-liver axis, including dysbiosis and increased gut permeability, may have an impact on NAFLD onset. Moreover, the impact of short-chain fatty acids (SCFAs) and succinate (microbial metabolites) on the microbiome promotes improvement in obesity-related insulin sensitivity in animals and influences mass control and glucose and lipid homeostasis in humans [74,75,76]. SCFAs, butyrate in particular, play a major role in maintaining the integrity of the intestinal wall, which prevents the migration of toxic and proinflammatory molecules into the liver [77]. Studies in rodent models of NAFLD fed a high-fat diet showed that sodium butyrate protected against the onset of fatty liver and decreased the IR [78,79]. Another well-known microbiome-derived compound is lipopolysaccharide, which is released in the colon from dead Gram-negative bacteria. Lipopolysaccharide plays a key role in the progression of metabolic diseases, such as NAFLD, insulin resistance and T2DM [80].

Obesity-induced IR is important in NAFLD and T2DM, and research findings show that IR may also be regulated by the gut microflora. The composition of the microbiome of obese individuals with T2DM is noticeably different from that of healthy people [81]. Men with MetS and IR, after receiving intestinal microbiota infusions from lean individuals, had a higher amount of butyrate produced and showed an improvement in insulin sensitivity [82]. Moreover, in patients with T2DM, a high-fat diet may lead to increased intestinal permeability and inflammation [83,84].

There is a significant link between NAFLD and cardiovascular disease. Fatty liver is associated with a high degree of calcification of the coronary arteries, regardless of the presence of metabolic syndrome, and risk factors for cardiovascular disorders and metabolic syndrome [85]. NAFLD patients have an increased risk of acute coronary syndrome and ischemic stroke due to endothelial dysfunction [86]. In the case of hepatic steatosis, an increase in arterial stiffness, a marker of cardiac hypertrophy and atherosclerotic lesions is also observed [87]. A meta-analysis involving 37 studies of metabolic syndrome confirmed the increased risk of cardiovascular events in this group of patients [88]. Another meta-analysis of 16 studies showed that a more severe course of NAFLD significantly increases the frequency of cardiovascular events [89]. Moreover, patients with fatty liver and coexistent T2DM have an almost four times higher risk of cardiovascular disease [85].

Moreover, recent studies have shown that treatment with direct-acting antivirals on HCV resulted in a reduction in HOMA-IR, an indicator that measures IR levels, in HCV patients [90]. Furthermore, the incidence of cardiovascular events was also reduced in both prediabetic and non-diabetic HCV patients treated with direct acting antivirals [91,92]. This suggests a link between liver disease and problems with the cardiovascular system.

An increase in fat content, which reduces insulin sensitivity, leads to increased lipolysis and an increase in FFA levels [93,94]. Moreover, skeletal muscle cells with IR have been shown to transfer stored glycogen to the de novo lipogenesis pathway in the liver [95,96]. Peripheral IR modifies lipid metabolism, which facilitates the generation of hepatic IR. Both visceral and peripheral fat content induce IR in hepatocytes [97]. Due to the hypertriglyceridemia present in fatty liver, visceral adipose tissue increases the production of leptin, which impairs insulin sensitivity, and reduces the level of adiponectin, which stimulates the action of insulin in peripheral tissues [98]. The decreased level of adiponectin leads to decreased antioxidant effects and the formation of hepatic fibrosis [99,100]. Recent studies showed an association between NAFLD and the incidence of fatty pancreas, also called nonalcoholic fatty pancreatic disease (NAFPD). NAFPD occurs in approximately 16% of the general adult population [101]. It is characterized by excessive accumulation of lipids in the pancreas [101,102]. Accumulated fat, and thus the presence of 12/15-lipoxygenase, lead to inflammation in pancreatic islet cells. Moreover, the accumulated fat may interfere with exocrine pancreatic function, which may lead to the development of pancreatic cancer [101]. There is also evidence of an association between fatty pancreatic and hepatic steatosis, and it can be argued that the presence of both conditions worsens the condition of NAFLD patients [103]. It has been shown that MetS risk factors are correlated with the risk of pancreatic steatosis, which may be a significant manifestation of MetS [104].

NAFPD is closely related, not only to MetS, but also to obesity, which is a risk factor for fatty pancreas. As for liver steatosis, there is also no approved treatment for pancreatic steatosis, but weight loss and bariatric surgery reduce the amount of fat in the pancreas [103]. Pancreatic steatosis occurs in approximately half of NAFLD patients, as confirmed by ultrasound, proton magnetic resonance spectroscopy and histopathological findings [105,106,107]. Human studies suggest that NAFPD also has an influence on the development of T2DM and prediabetic state [103]. Accumulated excess fat leads to loss of β-cell function, which may lead to diabetes [108]. In addition, NAFLD and NAFPD affect glucose metabolism and the function of β cells [102]. Prediabetic patients had a higher pancreatic fat content, but the relationship between insulin secretion and pancreatic fatty tissue has not been confirmed. However, there is evidence of the importance of IR in fatty pancreas [108]. Although a link with impaired sugar metabolism can be confirmed, the role of pancreatic fatty in the development of diabetes requires further clarification.

Mitochondria, responsible for beta-oxidation, are very important for metabolic disorders, NAFLD, and T2DM [109]. Mitochondrial dysfunction, and thus impaired fatty acid oxidation processes, leads to an increase in ROS levels and, consequently, to elevated oxidative stress [110]. ROS stimulate the activity of signaling pathways capable of inducing necroinflammation in liver cells. Impaired lipid beta-oxidation also leads to the accumulation of lipotoxic intermediates, which further increase inflammation and alter insulin signaling [111]. Insulin is important to mitochondria, due to their maintaining an appropriate NAD+/NADH ratio, while free radicals from mitochondria alter insulin sensitivity, disrupt insulin signaling and result in IR [112]. Molecules affecting the proper functioning of mitochondria include Slc25a1, associated with the metabolic processes of FFA and glycolytic pathways. Inhibition of Slc25a1 protects against NASH and reduces steatosis and steatohepatitis [113]. Another example is carnitine, which transports fatty acids into the mitochondrial matrix. It inhibits oxidative stress, enhances β-oxidation, and reduces IR. Carnitine supplementation improves both HOMA-IR and AST, ALT, and TG parameters in NAFLD patients [114].

## 4. Diagnosis

NAFLD is a common, often asymptomatic, liver disease. Therefore, its diagnosis requires the exclusion of other causes of steatosis, such as alcohol consumption [17]. Patients with NAFLD may report fatigue, daytime sleepiness, right upper quadrant abdominal pain or discomfort. On physical examination hepatomegaly and other signs of liver insufficiency are often present. In some cases, there are extrahepatic symptoms, such as joint and muscle pains [115].

Until now, steatosis required to be diagnosed by liver biopsy, which was considered the gold standard. Due to the fact that biopsy is an invasive method, transient elastography (TE), also known as FibroScan, is preferable. Unfortunately, biopsy is still considered the most accurate way to identify the stages of steatosis and fibrosis, despite the invasiveness of this method [116]. The most common method for identifying NAFLD is USS, because it is more widely available and cheaper than the other techniques. The USS diagnostic method has a sensitivity of 60–94% and a specificity of 66–95% [117]. However, this method does not reliably detect steatosis when it is below 20% in individuals with high BMI > 40 kg/m^2^ and is operator dependent [118].

CT (computer tomography) is non-invasive, widely available and easy to perform, but it also has some disadvantages, including a potential ionizing radiation hazard and limited accuracy in diagnosing mild hepatic steatosis [119].

Magnetic resonance imaging has numerous advantages, including being unaffected by obesity, simple steatosis, inflammation or the presence of ascites. On the other hand, these imaging methods are limited by high cost and low accessibility. They are also dependent on patient factors, such as inability to perform breath-hold, and signal degradation in patients with severe iron overload [120]. Nevertheless, some studies have shown, that MRI-PDFF (magnetic resonance imaging–proton density fat fraction) can be as effective as liver biopsy in steatosis diagnostics [121,122]. Apart from this, a quantitative estimation of liver fat can only be obtained by magnetic resonance spectroscopy (MRS). Therefore, this technique is of value in clinical trials and experimental studies, but is expensive and not recommended for everyday clinical practice [123].

Another imaging technique, the controlled attenuation parameter (CAP) using TE, can diagnose steatosis and fibrosis, but has limited ability to discriminate histological grades [124]. CAP has shown excellent diagnostic accuracy for detecting hepatic steatosis, in comparison to liver biopsy [125,126,127,128] and MRI-PDFF [129,130], even in morbidly obese individuals [131].

Liver enzyme activity (ALT, AST, GGT), total bilirubin, lipids fractions, apolipoprotein A1 (ApoA1), α-2-macroglobulin (α2M), haptoglobin (Hp), fasting blood glucose, and fasting insulin are used to generate liver steatosis scores (Table 2.). The best-validated scores are the fatty liver index (FLI), the SteatoTest and the NAFLD liver fat score [132,133]. They predict metabolic, hepatic and cardiovascular outcomes/mortality. These scores are associated with IR and reliably predict the presence, but not the severity, of steatosis [120].

Attenuation coefficient (AC) measurements using ultrasound waves have shown the ability to differentiate the severity of hepatic steatosis [134]. Furthermore, AC measurements have been found to be as effective as MRI-PDFF, even in obese people. However, the accuracy of the AC is limited by the occurrence of fibrosis and inflammation [135]. A similar technique is use of the backscatter coefficient (BSC), which has shown similar effects to those of AC [136].

For every NAFLD patient, surrogate markers of fibrosis (NFS, FIB-4, ELF or FibroTest) should be calculated. If significant fibrosis cannot be ruled out, patients should be referred to a liver clinic for further diagnostic procedures, e.g., TE, or, less frequently, magnetic resonance elastography (MRE), acoustic resonance force impulse (AFRI), or supersonic shearwave imaging (SSI) [120].

The final diagnosis should be made by liver biopsy. Liver biopsy is the only procedure that reliably differentiates NAFLD from NASH. Liver biopsy is an expensive and invasive procedure and may cause numerous complications, such as pain, bleeding, infection and, in some rare cases, increases mortality risk [132]. Additionally, its interpretation is influenced by the subjective judgment of the pathologist. It explores only a small portion of the liver parenchyma, which is not always representative of the entire parenchyma [137].

The prevalence of NAFLD is higher in both T2DM and prediabetes patients. In both conditions, the severity of NAFLD, progression to NASH, advanced fibrosis, and the development of HCC, can be observed independently of the level of liver enzymes [9,10]. According to recommendations for patients with T2DM, routine screening for NAFLD is not advised at this time because of uncertainties surrounding diagnostic tests and treatment options, along with lack of knowledge related to the long-term benefit and cost-effectiveness of screening [2].

Conversely, USS-defined NAFLD is associated with a 2–5-fold risk of developing T2DM, after adjustment for several lifestyle and metabolic confounders [8]. In persons with NAFLD, screening for diabetes is mandatory, by fasting or random venous glucose concentration, HbA1c concentration, or standardized 75 g oral glucose tolerance test (OGTT) [138].

A diagram demonstrating non-invasive diagnostics methods of NAFLD is presented below (Figure 2).

## 5. Complications

It seems that NAFLD is underdiagnosed in daily medical practice, even in patients with T2DM, even though coexistence of these pathologies increases the risk for patients. In subjects with NAFLD, T2DM appears to be a significant risk factor for advanced fibrosis. Additionally, not only T2DM, but also prediabetes are independently associated with portal inflammation, fibrosis, NASH and more severe histological findings in NAFLD patients [140].

Stepanova et al. found that patients with NAFLD and T2DM are at the highest risk for overall and liver-related mortality [141]. Of concern, T2DM is associated with a two-fold risk of chronic liver disease secondary to NAFLD, cirrhosis and HCC [17]. Moreover, family history of diabetes, especially among nondiabetics, is associated with NASH and fibrosis in NAFLD patients [142]. Furthermore, a diagnosis of NAFLD in patients with established T2DM is strongly associated with poor glycemic control, proliferative retinopathy, increased prevalence of cardiac/kidney disease and a 2.2-fold increase in all-cause mortality compared to subjects without NAFLD [8]. Hyperinsulinemia and hyperglycemia, and especially glycemic variability, are important predictive factors for the progression of hepatic fibrosis in NAFLD [143]. Lv et al. found that NAFLD was positively correlated with BMI, waist circumference (WC), TG, fasting blood glucose, diastolic blood pressure, and systolic blood pressure, but negatively correlated with the duration of diabetes, diabetic nephropathy, diabetic peripheral neuropathy and diabetic retinopathy and level of HDL-cholesterol [144].

The presence of NAFLD in T2DM patients may also increase the risk of cardiovascular disease (CVD) independently of MetS. Turkish researchers compared the CVD risk in T2DM and non-diabetic participants to evaluate the association between NASH and CVD risk. In this study 55 T2DM (study group) and 44 non-diabetic patients (control group) were included. Patients were also differentiated by the degree of hepatosteatosis. Hepatosteatosis rates were found to be similar in both diabetic and non-diabetic patients. Mean carotid intima–media thickness as cardiovascular risk assessment was found to be similar in both hepatosteatosis groups but substantially higher in diabetic patients, regardless of the degree of hepatosteatosis. Mean FPG and HbA1c were found to be higher in the grade ≥ 2 hepatosteatosis group [145].

A Polish study examined cardiovascular risk factors associated with NAFLD and the association between this pathology and macroangiopathy in T2DM patients. A total of 101 patients with T2DM were included in the study. Patients with NAFLD were significantly older but, surprisingly, the duration of diabetes was shorter. Patients with NAFLD had a statistically higher prevalence of coronary angioplasty, but there was no difference in the incidence of coronary heart disease and by-pass surgery. Significantly higher values of cardiovascular risk markers, such as HDL-cholesterol, ALT, and lower concentrations of creatinine were also found in this group of patients. Logistic regression analysis demonstrated that NAFLD was positively correlated with WC above normal and ALT activity but was negatively correlated with creatinine concentration. Further analysis showed that WC and total cholesterol were positive predictors of NAFLD and HDL-cholesterol was a negative prognostic parameter [146].

Targher et al. determined the prevalence of CVD and its risk factors between people diagnosed with T2DM with and without NAFLD. A total of 2839 patients with T2DM were screened. NAFLD was diagnosed by USS after exclusion of secondary causes of hepatic steatosis. To determine the risk of CVD, patients’ history, electrocardiogram, echo-Doppler scanning of lower limb and carotid arteries were used. Nearly 70% of patients had NAFLD, and this pathology was the most common liver disease. Its incidence increased with age. It was found that NAFLD patients had a higher prevalence of coronary, cerebrovascular and peripheral vascular disease than their counterparts without NAFLD, after adjustment for age and sex. Additionally, statistical analysis showed that NAFLD was associated with incidence of CVD independently of classical risk factors (e.g., age, sex, smoking, BMI, duration of diabetes, HbA1c, LDL-cholesterol), and actual use of medications, such as, antihypertensive, lipid-lowering, hypoglycemic or antiplatelet drugs [147]. Idilman et al. used coronary computed tomography angiography (CTA) to diagnose coronary artery disease (CAD) in T2DM patients. They also found that tomography-diagnosed NAFLD was associated with significant CAD, even after adjusting for age, gender, obesity, hypertension, smoking status and serum LDL-levels [148].

Bonapace et al. evaluated 50 patients with T2DM (32 patients with NAFLD and 18 without USS signs of fatty liver). Neither ischemic nor valvular heart disease was previously recognized in these patients, who underwent detailed examination by Doppler echocardiography, 24-h Holter monitoring and bicycle ergometry. They showed that presence of NAFLD may impair active and passive left ventricular diastolic function. Additionally, early LV diastolic function impairment in this subgroup of patients was independent of diabetes duration, HbA1c and other cardiometabolic risk factors (including age and sex) [149]. The same main authors, with colleagues, found a positive and independent association between NAFLD and aortic valve sclerosis (AVS) in patients with T2DM [150].

Targher et al. observed 400 patients with T2DM with no history of atrial fibrillation (AF) prospectively for 10 years. A total of 70.2% of these patients had ultrasounds signs of NAFLD. Each year, for every patient, standard 12-lead electrocardiogram (ECG) was performed. During this time, they found 42 incidents of AF. It is important to highlight that 90% of the patients had NAFLD at baseline. Additionally, patients with NAFLD had higher systolic and diastolic blood pressures and pulse pressure. The incidence of AF substantially increased after six years of observation. Statistical analysis revealed that, after adjustment for sex, age, hypertension and ECG changes (PR interval and left ventricular hypertrophy), NAFLD was associated with higher incidence of AF interval [151].

NAFLD not accompanied by IR is not associated with carotid atherosclerosis burden. However, having both NAFLD and IR seems to be an independent predictor of increased C-IMT (carotid intima-media thickness test) [152]. Moreover, the presence of NAFLD in patients admitted for acute ischemic stroke does not appear to be associated with more severe stroke or with worse in-hospital outcomes [153].

In T2DM patients aged 60–74 years both sex (with and without secondary causes of hepatic steatosis), the presence of USS signs of hepatic steatosis was not associated with reduction in renal function based on glomerular filtration rate and albuminuria during a 4-year follow-up [154]. Kim et al. found that NAFLD is inversely associated with prevalence of diabetic retinopathy and nephropathy in Korean patients with T2DM [155].

T2DM that coexists with NAFLD elevates the risk of cirrhosis and HCC. These patients have dyslipidemia, as well as higher hepatic steatosis and inflammation. Moreover, they demonstrate increased blood pressure, LDL and TG, while their HDL level is lower. Thus, T2DM patients with NAFLD have more severe dyslipidemia, hyperinsulinemia and hepatic IR than those without fatty livers [156].

It seems that a diagnosis of NAFLD is associated with a low risk of complications. The risk of complications, mainly cardiovascular, dramatically increases when NAFLD is accompanied by T2DM.

## 6. NAFLD Pharmacotherapy with Hypoglycemic Agents

According to the recommendations of the AASLD, the management of patients with NAFLD requires not only treatment of liver disease, but also of comorbidities, such as obesity, IR, lipid disorders and T2DM. Although, NAFLD has been researched for decades, there is no approved pharmacological treatment for the disease. Due to this, proper diet and increased physical activity are recommended.

The basic form of treatment for patients with NAFLD and T2DM is lifestyle modification. A sedentary lifestyle is observed not only in patients with hepatic steatosis, but also in those with metabolic syndrome and T2DM [157]. Al-Jiffri et al. found that an approximate 15% reduction in BMI (aerobic exercise training and diet) is effective in improving liver condition and insulin resistance in T2DM patients with NAFLD (i.e., reduction in ALP, ALT, AST, GGT and HOMA-IR) [158]. In NAFLD patients, aerobic exercise improved markers of hepatocyte function and insulin sensitivity regardless of any change in body weight [159]. Generally, people with a healthy lifestyle are less prone to develop IR, impaired glucose tolerance and diabetes [160,161]. According to the AASLD guidance document, exercise can prevent or improve hepatic steatosis in NAFLD, and reduce the likelihood of NASH. Furthermore, vigorous activity has more benefits for NASH patients than moderate activity [2].

A special diet is an important part of treatment of NAFLD and T2DM. Bozzetto et al. suggest that an isocaloric diet enriched in monounsaturated fatty acids (MUFA) compared with a diet higher in carbohydrate and fiber was associated with a clinically significantly lower hepatic fat content in T2DM patients independently of pursuit of an aerobic training program. They suggest that this diet should be considered for nutritional management of hepatic steatosis in people with T2DM [162].

Time-restricted feeding (TRF) is a dietary approach in which access to food is available for 8 h and unavailable for 16 h per day. TRF significantly reduced weight in NAFLD patients [163] and the severity of hepatic steatosis and hyperinsulinemia in mice [164]. Unfortunately, nonpharmacological management of NAFLD and effective weight loss pose some problems, since Stewart K et al. demonstrated that only 10.4% of overweight/obese individuals with NAFLD changed their habits [165].

Clinical trials examining the pharmacotherapy of NAFLD have focused mainly on insulin sensitizers, however, the data is scarce, as the number of studies evaluating the efficacy of glucose lowering agents in patients with NAFLD is small [166].

### 6.1. Metformin

Data from clinical trials which assessed the usefulness of metformin in the treatment of NAFLD are not consistent. In some cases, metformin in T2DM patients reduced transaminase levels and histological damage [167]. According to the majority of studies, metformin leads to a significant loss in IR and weight reduction in patients with NAFLD [168]. In contrast, Haukeland et al., in a small randomized trial (*n* = 48), reported that metformin compared with placebo over six months of treatment did not improve liver histology in patients with NAFLD [169]. In animal models with hyperphagic OLETF, aerobic exercise training of rats was more effective than metformin administration in the management of T2DM and NAFLD outcomes. Combining therapies offered little additional benefit beyond exercise alone, and findings suggest that metformin potentially impairs exercise-induced hepatic mitochondrial adaptation [170].

### 6.2. Thiazolidinediones (TZDs)

TZDs are the most potent evidence-based drugs against NASH [171]. In a rat model, pioglitazone and rosiglitazone prevented activation of hepatic stellate cells in vitro and improved hepatic steatosis and fibrosis [172]. Shadid et al. showed, that pioglitazone improved liver function in obese volunteers with NAFLD [173]. In a Polish study, NAFLD treatment with rosiglitazone was associated with significant improvement in liver enzyme activity and insulin sensitivity. It should be underlined that this therapy was very safe and well tolerated by patients, without adverse effect on lipid metabolism [174]. A systematic review of the value of insulin sensitizers in patients with NAFLD showed that pioglitazone improves all parameters of liver histology [168]. However, after the discontinuation of treatment, transaminase levels may return to baseline values [175]. A meta-analysis evaluating the effects of pioglitazone treatment for patients with NAFLD combined with T2DM demonstrated significant improvement in steatosis, ballooning and inflammation but had no effect on fibrosis [176].

The efficacy of rosiglitazone on NAFLD was observed in a FLIRT study, which revealed a significant antiestrogenic effect in the first year of therapy, although there was no additional benefit with longer-term treatment [177]. In a study [178], 64 patients with both NAFLD and impaired glucose metabolism (impaired glucose tolerance or T2DM) were treated with metformin 1700 mg/day, rosiglitazone 4 mg/day or combined therapy for 12 months. After that time, BMI decreased for patients given metformin and a combination of drugs (*p* = 0.002, *p* = 0.006, respectively). Moreover, postprandial glucose was reduced in all three groups, and liver biopsy showed improved NAFLD activity scores after treatment with rosiglitazone (*p* = 0.01) and combined therapy (*p* = 0.03).

The second-generation insulin sensitizer, MSDC-0602K has been shown to reduce ballooning and inflammation with no fibrosis progression after 12-months of treatment. Moreover, treatment with MSDC-0602K did not show the side effects observed with first-generation insulin sensitizers [179].

### 6.3. GLP-1 Agonists and DPP-4 Inhibitors

Incretin-mimetics, such as exenatide and liraglutide, have generated great interest because of their potential to reduce hepatic steatosis in patients with NAFLD. The impact of exenatide and liraglutide therapy during six months treatment of obese, T2DM patients with hepatic steatosis on intrahepatic lipid (IHL) was explored by Cuthbertson et al. [180]. They determined the relationship between changes in IHL with HbA1c, body weight, volume of abdominal visceral and subcutaneous adipose tissue (VAT and SAT). IHL was measured by liver proton MRS (^1^H MRS), while VAT and SAT were measured by whole body MRI. Patients were previously treated for at least three months with maximal tolerated doses of metformin, with either sulfonylureas (SU) or DPP-IV inhibitors. After conversion to GLP-1 agonists (exenatide, *n* = 19; liraglutide, *n* = 6), the following were observed: weight reduction (5.0 kg), 1.6% decrease in HbA1c, decrease in mean value of ALT activity (from 40 to 31 U/L) and GGT (from 69 to 43 U/L), 42% relative decrease in IHL and 7–11% in abdominal SAT or VAT. It should be emphasized that most of the patients had normal values of liver enzymes which significantly correlated with IHL. Compared with metformin, exenatide was not only found to better control blood glucose and body weight, but also to improve hepatic enzymes, attenuating hepatic steatosis in patients with T2DM concomitant with NAFLD [181].

Exenatide has a better hepatic-protective effect than intensive insulin therapy and, perhaps, represents a unique option for adjunctive therapy for patients with obesity, NAFLD with elevated liver enzymes and T2DM [182].

The LEAD-2 (Liraglutide Effect and Action in Diabetes) study examined the effect of liraglutide used for 24 weeks on liver function. In this placebo-controlled study, the effect of the drug was assessed by means of CT. It was found that 50.8% of patients with T2DM had an abnormal ALT at baseline. Application of 1.8 mg of liraglutide significantly decreased ALT activity and hepatic fat level compared to placebo, the effect being dose-dependent. Unfortunately, this effect was not observed after taking into account reduction in weight and HbA1c. In the liraglutide group, patients with and without baseline abnormal ALT did not differ in frequency of adverse effect. However, patients treated with liraglutide showed an improvement in hepatic steatosis—this trend also disappeared after adjusting for weight and HbA1c [183]

Sitagliptin used in T2DM coexisting with NAFLD was found to be safe and showed a similar antidiabetic effect as reported in patients with T2DM without NAFLD. It seems that tight glycemic control may contribute to the improvement of NAFLD. This observation was based on the correlation between changes in HbA1c and transaminases levels [184]. On the other hand, another study found no difference between various doses of sitagliptin (50–100 mg) on AST and ALT levels in comparison to physical activity and diet, although sitagliptin did decrease HbA1c after 1 year of treatment [185].

In a retrospective study of 82 patients with NAFLD and T2DM, Japanese researchers compared the efficacy of liraglutide, sitagliptin and pioglitazone. In all treatment groups they found significant improvements, not only in HbA1c level, but also in fasting blood glucose and ALT. Groups treated with liraglutide or pioglitazone had significantly lower AST to platelet ratios. Body weight was significantly lower in the liraglutide group (81.8 kg to 78.0 kg) and statistically higher in the pioglitazone group. This parameter did not change in the sitagliptin group. Multivariate regression analysis suggested that administration of liraglutide is an independent factor affecting body weight reduction of more than 5% [186]. In accordance with ADA or EASD statements, pioglitazone seems to be the preferred drug for patients with NAFLD and T2DM. Unfortunately, pioglitazone is not recommended by these institutions in cases of active liver disease and where liver enzymes are 2.5 times higher than the upper limit. It is emphasized that the use of PSU is rarely associated with an increase in liver enzymes, and their increase is not a contraindication to TZDs. The same prescribing guidelines are pertinent to meglitinides. In cases of severe liver damage, secretagogues should not be used because of the increased risk of hypoglycemia. Incretin-based drugs can be used safely in patients with NAFLD. EASD and ADA have emphasized that the limitations, particularly in patients with a history of pancreatitis, should be remembered. Fatty liver does not constitute a contraindication for insulin therapy, and for some patients with T2DM and NAFLD is the only therapeutic option [187].

### 6.4. Others

Lipid lowering medications (e.g., statins, fibrates, proprotein convertase subtilisin/kexin type 9), antioxidants (e.g., vitamin E, pentoxifylline and alpha lipoid acid), colesevelam, drugs that induce weight loss (e.g., orlistat and sibutramine), deferoxamina (an iron-chelating agent), antibiotics, probiotics, rimonabant, telimisartan, melatonin, betaina, aramchol, ursodeoxycholic acid, adenosine receptor agonists, nitro-aspirin, and omega-3 supplements, have all been considered in the treatment of NAFLD [137,174,188].

Bariatric surgery is an effective treatment for obesity and has been shown to markedly improve and even cure diabetes [89,90,189]. It also leads to improvement in histological features, ALT, AST and GGT levels, as well as lipid metabolism and inflammation [190,191,192]. The exact mechanisms that lead to improvement in NAFLD following bariatric surgery are not completely understood [193].

## 7. Summary

Coexistence of NAFLD and T2DM is common in everyday outpatient practice. NAFLD carries a low risk of complications, but the coexistence of NAFLD and T2DM significantly worsens prognosis. In persons with NAFLD, screening for diabetes is obligatory; on the other hand, in patients with diabetes, screening for NAFLD is not currently recommended. In treatment of NAFLD patients with diabetes, some hyperglycemic agents are useful, but lifestyle modification has the highest effectiveness. Further studies are needed because there is still a lack of adequate evidence-based methods for NAFLD screening. Assessment of the impact of newer anti-diabetic treatments, and identification of additional novel targets to treat NAFLD in diabetes patients, are also needed.

## Figures and Tables

**Figure 1 jcm-11-01375-f001:**
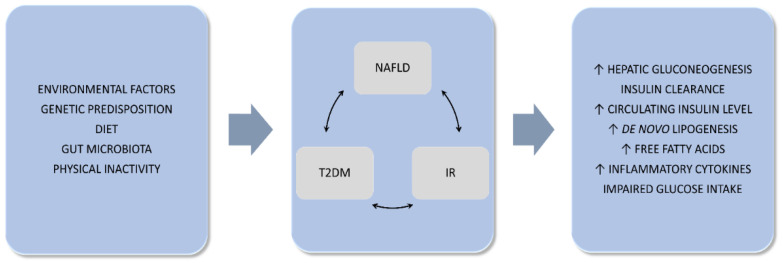
Common pathophysiologic mechanism in nonalcoholic fatty liver disease and type 2 diabetes mellitus. IR–insulin resistance; NAFLD–nonalcoholic fatty liver disease; T2DM–type 2 diabetes mellitus. The up arrow means height.

**Figure 2 jcm-11-01375-f002:**
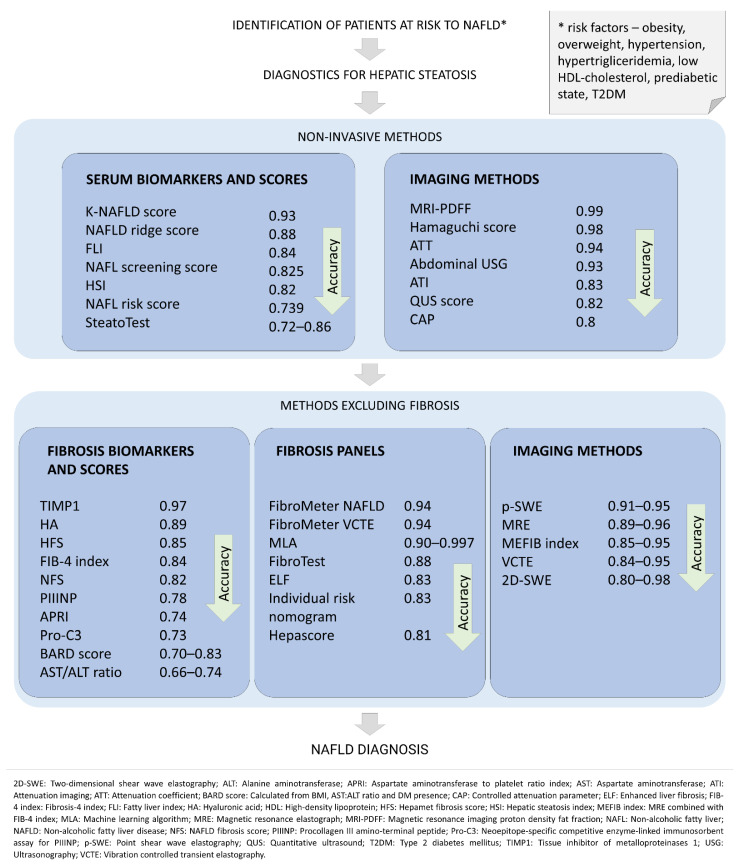
Diagram demonstrating the diagnosis methods of NAFLD in order of accuracy. Accuracy is presented as AUC (area under the receiver-operating characteristics curve) value. The figure does not include liver biopsy, which remains the gold standard procedure for the diagnosis of nonalcoholic steatohepatitis (NASH) and staging of nonalcoholic fatty liver disease (NAFLD) [139].

**Table 1 jcm-11-01375-t001:** Secondary causes of liver steatosis.

Medications	cART in HIV, chemotherapy, amiodarone, methotrexate, tamoxifen, corticosteroids, tetracyclines, valproic acid, amphetamines, acetylsalicylic acid
Genetic causes	haemochromatosis, alpha-1 antitrypsin deficiency, Wilson’s disease, congenital lipodystrophy, abetalipoproteinaemia, hypobetalipoproteinaemia, familial hyperlipidaemia, lysosomal acid lipase deficiency, glycogen storage diseases, hereditary fructose intolerance, urea cycle disorders, citrin deficiency
Environmental causes	lead, arsenic, mercury, cadmium, herbicides, pesticides, polychlorinated biphenyls, chloroalkenes
Nutritional/gastroenterological causes	severe surgical weight loss, starvation, malnutrition, total parenteral nutrition, microbiome changes, coeliac disease, pancreatectomy, short bowel syndrome
Other causes	Chronic HCV infection, hypothyroidism, polycystic ovary syndrome, hypothalamic or pituitary dysfunction, growth hormone deficiency, HELLP syndrome, acute fatty liver of pregnancy, celiac disease, Wilson’s disease, hepatitis C virus, *Amanita phalloides* mushrooms, phosphorous poisoning, petrochemicals, *Bacillus cereus* toxin

cART—combined antiretroviral therapy; HIV—human immunodeficiency virus; HELLP—hemolysis, elevated liver enzymes and low platelets.

**Table 2 jcm-11-01375-t002:** Non-invasive scoring system to diagnose NAFLD progression.

Index	Factors
AST/ALT ratio	AST, ALT
APRI (AST to platelet ratio index)	AST, upper normal limit for ALT, PLT
FibroTest	Age, gender, total bilirubin, haptoglobin, GGT, α2-macroglobulin, apolipoprotein-A
FibroMax	Age, gender, total bilirubin, haptoglobin, GGT, α2-macroglobulin, apolipoprotein-A, ALT, AST, TCH, TG, fasting glucose, weight, height
FibroMeter	HA, PLT, prothrombin index, α2-macroglobulin
BARD	BMI, AST/ALT ratio, diabetes mellitus
NFS (NAFLD fibrosis score)	Age, hyperglycemia, BMI, PLT, albumin, AST/ALT ratio
FIB-4 (fibrosis 4 index)	Age, AST, ALT, PLT
HepatoScore	Age, gender, bilirubin, GGT, HA, α2-macroglobulin
OELF (Original European Liver Fibrosis panel)	Age, TIMP 1, HA, P3NP
ELF (European Liver Fibrosis panel)	HA, P3NP, TIMP-1
NIKEI (Noninvasive Koeln–Essen-index)	Age, AST, AST/ALT ratio, total bilirubin

ALT–alanine aminotransferase; AST–aspartate aminotransferase; BMI–body mass index; GGT–gamma-glutamyltransferase; HA–hyaluronic acid P3NP–amino-terminal propeptide of type III procolagen; PLT–platelet count; TCH–total cholesterol; TG–trigycerides; TIMP-1–tissue inhibitor of metalloproteinase 1.

## Data Availability

Not applicable.

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
