# Peer review of "The Coexistence of Nonalcoholic Fatty Liver Disease and Type 2 Diabetes Mellitus"

_jcm, 2022, doi:10.3390/jcm11051375_

Round 1
Reviewer 1 Report
I enjoyed reading this narrative review on the association between NAFLD and T2DM. The manuscript is interesting. The topic is hot. Figures and tables are clear.
This reviewer strongly suggests adding and commenting some up-to-date references closely related to the topic to further enrich this review.
1- In type 2 diabetes mellitus, the border between NAFLD and NASH is very thin. In fact, it was recently observed by liver biopsy that steatohepatitis represents the sole feature of liver damage in type 2 diabetes. In other words, NAFLD generally presents as NASH in type 2 diabetic patients (PLoS One. 2017 Jun 1;12(6):e0178473. doi: 10.1371/journal.pone.0178473.). This intriguing point should be discussed in the review.
2- It is well known that IR is the strongest pathophysiological link between NAFLD and Metabolic Syndrome. Recent studies have shown that the reduction of IR through the pharmacological eradication of HCV by direct-acting antivirals leads to both a reduction in the onset of type 2 diabetes (Diabetes, Obesity and Metabolism, 2020, 22(12):2408–2416. doi: 10.1111/dom.14168) and clinical expressions of atherosclerosis (Atherosclerosis, 2020, 296:40–47. doi: 10.1016/j.atherosclerosis.2020.01.010 - Nutrition, Metabolism & Cardiovascular Diseases (2021) 31, 2345e2353. doi: 10.1016/j.numecd.2021.04.016). These interesting issues as well as the above references deserve to be commented in the text by the authors
3- The strict relationship between NAFLD and type 2 diabetes involves some pathophysiological mechanism still poorly studied or neglected today. In particular, I am referring to the role of opioid system, both on NAFLD (J Pharmacol Exp Ther. 2015 Mar;352(3):462-70. doi: 10.1124/jpet.114.220764.) and on insulin resistance/obesity (J Clin Endocrinol Metab. 1996 Feb;81(2):713-8. doi: 10.1210/jcem.81.2.8636293). This intriguing issue and above references should be added in discussion.
4- NAFLD and IR are bidirectionally correlated and, consequently, the development of pre-diabetes and diabetes is the most direct consequence at the extrahepatic level. Two very recent reviews explain in an updated and complete way the pathophysiological mechanisms that support the relationship between IR and NAFLD (Antioxidants, 2021, 10(2), pp. 1–25, 270. doi: 10.3390/antiox10020270. - Processes, 2021, 9(1), pp. 1–18, 135. doi: 10.3390/pr9010135), Authors should add the above references to the manuscript.
5- Mitochondrial dysfunction in NAFLD is an interesting novel issue (Biochim Biophys Acta Mol Basis Dis. 2020 Oct 1;1866(10):165838. doi: 10.1016/j.bbadis.2020.165838.). Actually, this issue should be addressed in the manuscript.
7- Finally, the pathophysiological mechanisms that link NAFLD with the progression of atherosclerotic damage and therefore with the risk of MACE have been extensively described in a very recent review (Rev Cardiovasc Med. 2021 Sep 24;22(3):755-768. doi: 10.31083/j.rcm2203082.).
Author Response
We thank for thoughtful review of our work. We carefully considered all comments and made necessary changes according to recomendations.
Point 1: I enjoyed reading this narrative review on the association between NAFLD and T2DM. The manuscript is interesting. The topic is hot. Figures and tables are clear.
This reviewer strongly suggests adding and commenting some up-to-date references closely related to the topic to further enrich this review.
In type 2 diabetes mellitus, the border between NAFLD and NASH is very thin. In fact, it was recently observed by liver biopsy that steatohepatitis represents the sole feature of liver damage in type 2 diabetes. In other words, NAFLD generally presents as NASH in type 2 diabetic patients (PLoS One. 2017 Jun 1;12(6):e0178473. doi: 10.1371/journal.pone.0178473.). This intriguing point should be discussed in the review.
Response 1: Modyfication to manuscript:
“Nevertheless, the border between NAFLD and NASH is very thin. It is difficult to differentiate simple fatty liver from steatohepatitis without using invasive diagnostic methods. In a study of T2DM patients, each of the 63 research participants had fatty liver, but 94.82% of them had NASH, which was confirmed by a liver biopsy. This suggests that steatohepatitis may be associated with the early stages of T2DM development, perhaps due to the association of NASH with IR [64].”
Point 2: It is well known that IR is the strongest pathophysiological link between NAFLD and Metabolic Syndrome. Recent studies have shown that the reduction of IR through the pharmacological eradication of HCV by direct-acting antivirals leads to both a reduction in the onset of type 2 diabetes (Diabetes, Obesity and Metabolism, 2020, 22(12):2408–2416. doi: 10.1111/dom.14168) and clinical expressions of atherosclerosis (Atherosclerosis, 2020, 296:40–47. doi: 10.1016/j.atherosclerosis.2020.01.010 - Nutrition, Metabolism & Cardiovascular Diseases (2021) 31, 2345e2353. doi: 10.1016/j.numecd.2021.04.016). These interesting issues as well as the above references deserve to be commented in the text by the authors
Response 2: Modyfication to manuscript:
“Moreover, recent studies have shown that treatment with direct acting antivirals on HCV resulted in a reduction in HOMA-IR, an indicator that measures IR levels, in HCV patients [108]. Furthermore, the incidence of cardiovascular events is also reduced in both prediabetic and non-diabetic HCV patients treated with direct acting antivirals [109]. This suggests a link between liver disease and problems with the cardiovascular system.”
Point 4: NAFLD and IR are bidirectionally correlated and, consequently, the development of pre-diabetes and diabetes is the most direct consequence at the extrahepatic level. Two very recent reviews explain in an updated and complete way the pathophysiological mechanisms that support the relationship between IR and NAFLD (Antioxidants, 2021, 10(2), pp. 1–25, 270. doi: 10.3390/antiox10020270. - Processes, 2021, 9(1), pp. 1–18, 135. doi: 10.3390/pr9010135), Authors should add the above references to the manuscript.
Response 4: Modyfication to manuscript:
“An increase in fat content, which reduces insulin sensitivity, leads to increased lipolysis and an increase in FFA levels [90,91]. Moreover, skeletal muscle cells with IR have been shown to transfer stored glycogen to the de novo lipogenesis pathway in the liver [92,93]. Peripheral IR modifies lipid metabolism, which facilitates the generation of hepatic IR. Both visceral and peripheral fat content induce IR in hepatocytes [94]. Due to the hypertriglyceridemia present in fatty liver, visceral adipose tissue increases the production of leptin, which impairs insulin sensitivity, and reduces the level of adiponectin, which stimulates the action of insulin in peripheral tissues [95]. The decreased level of adiponectin leads to decreased antioxidant effects and the formation of hepatic fibrosis [96,97].”
Point 5: Mitochondrial dysfunction in NAFLD is an interesting novel issue (Biochim Biophys Acta Mol Basis Dis. 2020 Oct 1;1866(10):165838. doi: 10.1016/j.bbadis.2020.165838.). Actually, this issue should be addressed in the manuscript.
Response 5: Modyfication to manuscript:
“Mitochondria, responsible for the beta-oxidation, are very important for metabolic disorders, NAFLD, and T2DM [102]. Mitochondrial dysfunction, and thus impaired fatty acid oxidation process, leads to an increase in ROS levels and, consequently, to elevated oxidative stress [103]. ROS stimulate the activity of signaling pathways capable of induc-ing necroinflammation in liver cells. Impaired lipid beta-oxidation also leads to the accu-mulation of lipotoxic intermediates, which further increases inflammation and alters insu-lin signaling [104]. Insulin is important to mitochondria, due to the maintaining the ap-propriate NAD+/NADH ratio, while free radicals from mitochondria alter insulin sensitiv-ity, disrupt insulin signaling and result in IR [105]. The molecules affecting the proper functioning of mitochondria include Slc25a1, associated with the metabolic processes of FFA and glycolytic pathways. Inhibition of Slc25a1 protects against NASH and reduces steatosis and steatohepatitis [106]. Another example is carnitine, which transports fatty acids into the mitochondrial matrix. It inhibits oxidative stress, enhances β-oxidation, and reduces IR. Carnitine supplementation improves both HOMA-IR and AST, ALT, TG pa-rameters in NAFLD patients [107].”
Point 6: Finally, the pathophysiological mechanisms that link NAFLD with the progression of atherosclerotic damage and therefore with the risk of MACE have been extensively described in a very recent review (Rev Cardiovasc Med. 2021 Sep 24;22(3):755-768. doi: 10.31083/j.rcm2203082.).
Response 6: Modyfication to manuscript:
“There is a significant link between NAFLD and cardiovascular diseases. Fatty liver is associated with a high degree of calcification of the coronary arteries, regardless of the presence of the metabolic syndrome and risk factors for cardiovascular disorders and metabolic syndrome [74]. NAFLD patients have an increased risk of the acute coronary syndrome and ischemic stroke due to endothelial dysfunction [75]. In the case of hepatic steatosis, an increase in arterial stiffness, a marker of cardiac hypertrophy and atherosclerotic lesions is also observed [76]. A meta-analysis involving 37 studies with metabolic syndrome confirms the increased risk of cardiovascular events in this group of patients [77]. Another meta-analysis of 16 studies showed that a more severe course of NAFLD significantly increases the frequency of cardiovascular events [78]. Moreover, patients with fatty liver and coexistent T2DM have an almost 4 times higher risk of cardiovascular diseases [74].”
We re greatful for your time and effort necessary to provide insightful comments.
Reviewer 2 Report
In the manuscript entitled “The coexistence of nonalcoholic fatty liver disease and type 2 diabetes mellitus”, the authors emphasize the importance of studying coexistence between NAFLD and T2DM. They first summarize the different causes for NAFLD, and the epidemiology of NAFLD, especially the epidemiology of NAFLD in diabetic patients. Then the authors describe in-depth the IR-induced pathogenesis of NAFLD by bringing up the “multi-hit hypothesis” and list all the possible diagnosis and their limitations. Lastly, they outline the complications of NAFLD and T2DM, which include cardiovascular diseases, atrial fibrillation, cirrhosis, and HCC, and outline drugs that can be beneficial to control the progress of the NAFLD other than health management.
Major comment:
- In terms of the different diagnosis methods of NAFLD, it would be better if the authors can make a diagram listing the diagnosis methods in the order of accuracy and effectiveness.
Minor comment:
- Grammar issue. In line 253, the word associated is misused. Original sentence: “Both 252 conditions are closely associate with the severity of NAFLD…”
- Text issue. Some of the words are highlighted in light blue, and some are underlined. Please browse through the main text and fix that.
Author Response
We thank for thoughtful review of our work. We carefully considered all comments and made necessary changes according to recomendations.
Point 1: In terms of the different diagnosis methods of NAFLD, it would be better if the authors can make a diagram listing the diagnosis methods in the order of accuracy and effectiveness.
Response 1: Modyfication to manuscript: diagram included in the manuscript.
Point 2: Grammar issue. In line 253, the word associated is misused. Original sentence: “Both 252 conditions are closely associate with the severity of NAFLD…”
Response 2: Modyfication to manuscript:
“In both conditions, the severity of NAFLD, progression to NASH, advanced fibrosis and the development of HCC can be observed, independently of the level of liver enzymes.”
Point 3: Text issue. Some of the words are highlighted in light blue, and some are underlined. Please browse through the main text and fix that.
Response 3: I have removed the underlines and replaced them with the following sentences:
“Liver enzymes activity (ALT, AST, GGT), total bilirubin, lipids fractions, apolipoprotein A1 (ApoA1), α-alpha 2-macroglobulin (α2M), haptoglobin (Hp), fasting blood glucose, fasting insulin are used to generate liver steatosis scores.”
“Turkish researchers compared the CVD risk in T2DM and non-diabetic participants to evaluate the association between NASH.”
We re greatful for your time and effort necessary to provide insightful comments.
Round 2
Reviewer 1 Report
Line 275-279. Addressing the second issue raised by this reviewer, the authors write “Moreover, recent studies have shown that treatment with direct acting antivirals on HCV resulted in a reduction in HOMA-IR, an indicator that measures IR levels, in HCV patients [90]. Furthermore, the incidence of cardiovascular events is also reduced in both prediabetic and non-diabetic HCV patients treated with direct acting antivirals [91]. This suggests a link between liver disease and problems with the cardiovascular system.”
Actually, the reference on the reduction of cardiovascular events in prediabetic patients is missing (Nutrition, Metabolism & Cardiovascular Diseases (2021) 31, 2345e2353. doi: 10.1016/j.numecd.2021.04.016). Authors need to add it.
Author Response
Thank the Reviewer’s very much for your time and valuable comments on our manuscript.
Point 1: Line 275-279. Addressing the second issue raised by this reviewer, the authors write “Moreover, recent studies have shown that treatment with direct acting antivirals on HCV resulted in a reduction in HOMA-IR, an indicator that measures IR levels, in HCV patients [90]. Furthermore, the incidence of cardiovascular events is also reduced in both prediabetic and non-diabetic HCV patients treated with direct acting antivirals [91]. This suggests a link between liver disease and problems with the cardiovascular system.”
Actually, the reference on the reduction of cardiovascular events in prediabetic patients is missing (Nutrition, Metabolism & Cardiovascular Diseases (2021) 31, 2345e2353. doi: 10.1016/j.numecd.2021.04.016). Authors need to add it.
Response 1: Thank the Reviewer’s suggestion. As suggested, we have included the references in the manuscript.
